# Evaluation of Hydrothermal Treatment of Winter Wheat Grain with Ozonated Water

**DOI:** 10.3390/plants12183267

**Published:** 2023-09-14

**Authors:** Simona Paulikienė, Renata Žvirdauskienė

**Affiliations:** 1Faculty of Engineering, Agriculture Academy, Vytautas Magnus University, Studentų Str. 15, 53362 Akademija, Lithuania; 2Department of Food Science and Technology, Kaunas University of Technology, Radvilėnų Str. 19, 50254 Kaunas, Lithuania; 3Microbiology Laboratory, Institute of Agriculture, Lithuanian Research Centre for Agriculture and Forestry, Instituto al. 1, 58344 Akademija, Lithuania

**Keywords:** ozonated water, moistening, grain, mould, contamination, quality

## Abstract

Products must be cleaned or otherwise treated to keep them clean when they are prepared for further production or when they are supplied fresh to the consumer. Cereals have significantly lower settling losses than succulent agricultural products, but the risks that can arise from their hydrothermal treatment before milling—where the cereals are moistened and left to rest for 14 h (temperature 30 °C)—are often underestimated. This operation creates a favourable environment for the development of micro-organisms, which, if not destroyed, can continue throughout the processing stages and be passed on to the consumer. This study investigated the qualitative characteristics of winter wheat hydrothermally treated with ozonated water at a concentration of 1.51 ± 0.1 mg L^−1^, such as the amount of mould in the grains and flour, as well as the grain protein, moisture, gluten, sedimentation, starch and weight per hectolitre. For the assessment of these parameters, the account was taken of the State standard, which provides the grain class and the type of grain. The reduction in mould fungi after the treatment of the winter wheat grain with ozonated water ranged between 440 and 950 CFU g^−1^. The results of the microbiological analysis showed that the ozone treatment improved the mycological safety of the flour samples made from the grain from an average of 390 ± 110 CFU g^−1^ to 29 ± 12 CFU g^−1^. On the other hand, the treatment of kernels with ozonated water did not significantly affect the basic composition of the winter wheat grains.

## 1. Introduction

Wheat is considered a dietary staple, a source of carbohydrates and protein, and plays an important role in meeting people’s energy and nutrient needs [1]. Wheat grown in fields is surrounded by a natural environment and is therefore inevitably exposed to many sources of micro-organism contamination, such as air, water, soil, insects, birds, rodents, etc. Inadequately managed biological, environmental, social and economic factors lead to increased crop losses, which can range between 15 and 53% at all levels of the food chain, depending on the product and the country [2,3]. Grain losses are around 24–34% across food chains [2], and as much as 15% of the grain grown in the world is lost due to microbiological contamination [4]. To reduce product waste, it is essential to improve food safety.

Although grain production loss is significantly lower compared to succulent products, the risk of microbiological contamination during hydrothermal treatment is often not assessed. After cleaning, the grain is hydrothermally treated, i.e., the grain mass is moistened with a certain amount of water and left to rest for a period of time to soften the inner endosperm and harden the bran [5]. This process aims to ensure that the bran gradually separates during milling [6]. This process provides the grain with the structural properties required for milling and facilitates the separation of the hull from the endosperm, as well as improving the efficiency of sieving. After wetting, the grain is stored in a warmer environment for a period of time [5]. The time and temperature of the hydrothermal treatment of grain generally vary depending on the type of grain, variety and initial moisture content [6]. For example, when winter wheat grains are moistened with water, their moisture content rises to 2% and they are stored for about 14 h. The wheat is exposed to a temperature of 30 °C for 14 h. This temperature is favourable for the growth of microorganisms. According to the FDA guidelines, it is considered to be relatively safe to consume grains when the micromycetes count of the product does not exceed 10^2^–10^4^ CFU g^−1^ [7]. Therefore, the lower the initial number of micromycetes in the grain, the safer the final product will be. Microorganisms that are not destroyed during the preparation processes reach the consumer “on the table”. Common methods of reducing micro-organisms are high temperature or chemical treatment. Unfortunately, these methods can adversely affect the chemical composition and quality of the grain [8,9]. Without preventive measures to reduce contamination, micro-organisms and the infections they cause travel through all of the processes and reach consumers. Preventing micro-organisms requires new methods to prevent or slow down their growth. At the same time, it is essential to ensure that the quality of processed cereals is stable in order to obtain the highest quality flour.

Ozone can be used to preserve crop products. This would help to address the growing concerns about the use of harmful pesticides and other chemicals to control pests [10]. Ozone is well-known for its ability to readily decompose to oxygen in the air, and on contact with the product, residues are minimal or they dissipate and there is no need to remove the resulting gas [11]. Ozone is an environmentally friendly, cost-effective and Generally Recognised as Safe (GRAS) food processing technology that kills 99% of microorganisms and is used instead of heat treatment as it does not require heat energy [11,12,13]. Thus, the ozone treatment of products can improve the economic benefits, microbiological safety and shelf life of quality food products.

Ozone gas is used to process cereal crops during drying, preparation for storage and storage. Ozone is also used in the pre-milling hydrothermal treatment of cereals, where the produce is moistened and left to rest in a gaseous ozone environment. This technology is patented and known worldwide as Oxygreen [14,15]. It is the treatment of grain with ozone in a closed pressurised reactor, which creates close, homogeneous and controlled contact between the gas and the grain [14]. During this operation, the grains are moistened with water and, in an environment saturated with ozone gas, the water is better absorbed by the grains and the separation of the grain parts is easier. However, the technology is still not commonly used. The most common practice is to moisten the grain with water and leave the grain to ‘lie flat’.

According to the literature, ozone has no effect [10,16,17,18] or a very small effect [19] on the quality parameters of cereal crops and improve other characteristics [17,18,20]. However, in many studies, ozone gas is the main preventive target used. Therefore, when ozonated water is used for grain humidification, it is very important to identify the positive and negative benefits of ozone.

The aim of the study was to investigate the influence of ozonated water on the contamination of winter wheat grain micromycetes during hydrothermal treatment, determining their effect on qualitative indicators.

## 2. Results

### 2.1. Evaluation of Micromycete Contamination of Winter Wheat Grain

During the research, observations based on morphological characteristics revealed that the most common microscopic fungi were *Alternaria*, *Cladosporium*, *Penicillium*, *Mucor*, *Aspergillus* and *Fusarium* microscopic fungi. All the tested grains were typically affected not by just one, but by several species of these mould fungi. As a result, there is a significant risk that these fungi will become active during storage and start producing toxins.

Firstly, a surface micromycete contamination test was carried out on winter wheat grains before and after the treatment to determine the number of mould fungi colony forming units (hereafter referred to as the mould fungi count). The results of the study are shown in Figure 1. In all cases (months), the average moisture content of the grain was 13.75 ± 0.36%, while after the hydrothermal treatment, the moisture content of the grain was 15.63 ± 0.46% with water humidification and 15.63 ± 0.46% with ozonated water humidification (*n* = 42).

The survey data show that there is a wide range of mould contamination. And there is a significant difference between the months (*p* < 0.05), meaning that the monthly averages cannot be compared. However, it can be seen that, in all cases (monthly results), the number of mould fungi colonies in the wheat humidified with ozonated water (1.51 ± 0.1 mg L^−1^) is significantly different both with and without humidification (except December)—i.e., the different letters in the month of the study show significant differences between the mean values, with a *p* value < 0.05 (according to the Tukey’s HSD test), in the case of the grains analysed with the ozonated water.

The first analysis of winter wheat grain at the company started in July, where the number of wheat mould fungi in the grain was 940.0 ± 70.2 CFU g^−1^. During the grain treatment with water, it was observed that the number of mould fungi decreased by 76.6% (220.0 ± 18.6 CFU g^−1^), and when treated with ozonated water, the number of mould fungi decreased by 93.6% (60.0 ± 3.8 CFU g^−1^). This was the highest change in the whole study. Another study was carried out in December, in which the surface mould contamination of the non-humidified grain was 110.0 ± 9.8 CFU g^−1^. This study recorded the highest increase both after humidification with water, which was—80.7% (570.0 ± 43.6 CFU g^−1^), and after humidification with ozonated water, which was—15.4% (130.0 ± 11.4 CFU g^−1^). However, the overall monthly assessment shows that the difference in the mould contamination of the grain humidified with ozonated water is 77.2% less than that of the grain humidified with water. In February and June, the contamination of the non-humidified winter wheat increased by 6.2% (760.0 ± 61.8 CFU g^−1^ → 810.0 ± 63.8 CFU g^−1^) and 17.9% (550.0 ± 42.0 CFU g^−1^ → 670.0 ± 50.6 CFU g^−1^) compared to the water-humidified winter wheat. In contrast, ozonated water reduced the contamination from the initial grain contamination by 61.8% (290.0 ± 22.2 CFU g^−1^) and 60.0% (220.0 ± 15.6 CFU g^−1^), respectively. The reductions recorded in March, April and May were among the highest, at 1100.0 ± 88.0 CFU g^−1^, 840.0 ± 65.2 CFU g^−1^ and 890.0 ± 70.2 CFU g^−1^, respectively. After water wetting, the number of mould fungi decreased by 15.5% (930.0 ± 71.4 CFU g^−1^), 4.8% (800.0 ± 63.0 CFU g^−1^) and 43.8% (500.0 ± 39.0 CFU g^−1^). On the other hand, the number of mould fungi was 83.9% (150.0 ± 11.0 CFU g^−1^), 69.4% (245.0 ± 17.6 CFU g^−1^) and 74.0% (130.0 ± 9.4 CFU g^−1^) lower in these months under ozonated humidification compared to water humidification.

The data showed that ozonated water at a concentration of 1.51 ± 0.1 mg L^−1^ reduced the mould contamination by an average of about 76.4% compared to the wheat before humidification, and about 72.8% times compared to the wheat humidified with water.

After the hydrothermal treatment, milling of the winter wheat into flour was carried out in a randomly selected month to investigate the effect of ozonated water on the micromycete contamination of the flour. Figure 2 shows the results of the analysis of the randomly selected samples of cereal flour, where the number of mould fungi in the flour was determined. The moisture content of the milled grain treated with water was 15.03 ± 0.06% and that of the grain treated with ozonated water was 15.05 ± 0.09%.

It was found that the number of mould fungi in the flour after hydrothermal treatment with ozonated water was reduced by an average of 92.6%. This indicates that it is possible to reduce the number of micromycetes that can move downstream in the production chain.

### 2.2. Evaluation of the Quality Parameters of Winter Wheat Cereals

The assessment included qualitative parameters such as the protein, gluten, sedimentation, starch and weight per hectolitre, i.e., the bulk density.

Figure 3A shows the protein content of the winter wheat grain both before and after humidification (13.53 ± 0.69%). Immediately after humidification, the protein content was as follows: wheat humidified with ozonated water—13.35 ± 0.71%; wheat humidified with water—13.37 ± 0.68%. After 14 h, the protein content in both cases was 13.60 ± 0.61% and 13.60 ± 0.58%, respectively.

Assessing the results of the study (significant at the *p* < 0.05 level, based on one-factor ANOVA with Tukey’s HSD test), it can be said that the change in proteins was not statistically significant.

The gluten content of the wheat (Figure 3B) decreased significantly only after humidification, both in the case of ozonated water (10.41%) and water (9.22%). However, the gluten content recovered after 14 h and was only slightly different from that of the non-humidified wheat. The gluten content of the ozonated wheat was 0.09% lower and 0.79% higher than that of the original wheat.

A one-factor ANOVA with Tukey HSD test at a *p* < 0.05 level of significance on the wheat gluten shows that, immediately after wetting, the gluten of the wheat is reduced in both cases compared to that of the non-wetted wheat, and that there is no significant difference between the initial and the 14-h wetted wheat levels.

The gluten content, according to the gluten characteristic of the standard LST 1524:2019 [21], is “strong”, as it falls within the following ranges: 20 to 40.

The wheat sedimentation rate (Figure 3C) increased immediately after humidification for both the ozonated and water-humidified wheat, which was 9.32% and 8.27% higher than the unhumidified wheat sedimentation rate, respectively. After 14 h, the sedimentation rate was reduced compared to the wheat immediately after humidification. It remained only slightly higher than that of the non-humidified group (0.66 and 0.05%). The sedimentation rates (ANOVA with Tukey HSD test, *p* < 0.05) show that, immediately after humidification, the sedimentation rate of the wheat is increased in both cases compared to the non-humidified wheat, while there is no significant difference between the sedimentation rates of the wheat at the start of the humidification period and those of the humidified wheat 14 h later.

Figure 3D shows the variation of the wheat starch after wetting and after 14 h. Immediately after humidification, the starch content of the wheat humidified with ozonated water and with water decreased by 1.60 and 1.20%, respectively, compared with that of the non-humidified wheat. Although the apparent variation in wheat starch appears to be marginal, both immediately after humidification and after 14 h, there are visible differences between the mean values of the indicators according to the one-factor ANOVA with the Tukey HSD test at a significance level of *p* < 0.05.

Looking at the weight per hectolitre of the wheat (ANOVA with Tukey HSD test at *p* < 0.05 level of significance), the highest value is observed for the un-humidified wheat (Figure 3E), with a significant difference between the wheat immediately after humidification and after 14 h. If the humidified wheat after 14 h is compared, there is no significant difference between the storage treatments. Immediately after humidification, the drop was the highest, i.e., 17.75% for the wheat humidified with ozonated water and 16.58% for the wheat humidified with water. After 14 h, it had risen and was lower than that of the non-humidified wheat, at 10.8% and 9.57%, respectively.

As can be seen, the 0.12% change in protein was lower after ozonated water humidification and levelled off after 14 h of storage. The gluten content of the wheat humidified with water was 1.31% (immediately after humidification) and 0.88% (after 14 h) higher than that of the wheat humidified with ozonated water, the starch content of the wheat was higher by 0.41% and 0.40%, respectively, and the weight per hectolitre was higher by 1.40% and 1.36%. However, the sedimentation rate of the wheat humidified with water was 1.15% (immediately after humidification) and 0.61% (after 14 h) lower than that of the wheat humidified with ozonated water. In conclusion, in all cases, the qualitative parameters were not significantly affected by the ozone preventive measure, i.e., no significant difference was found between these parameters, and the data presented above (Figure 1, Figure 2 and Figure 3) suggest that there is no significant difference between the treatments in the qualitative parameters of winter wheat. The moisture content of the wheat humidified with water was 0.65 and 1.28% higher than that of the wheat humidified with ozonated water. Thus, ozone did not increase the water uptake.

## 3. Discussion

### 3.1. Evaluation of Micromycete Contamination of Winter Wheat Grain

A review of the studies that have previously been carried out shows that more studies have been carried out on cereals using ozone gas, including the treatment of cereals before milling—i.e., first the introduction of water followed by ozonation in the environment (Oxygreen^®^). Allen et al. [22], investigating the use of ozone gas on barley, found that ozone has a greater effect on spores than on the mould fungi themselves, i.e., that spores and mixtures of spores with a small number of mould fungi colonies can be reduced by up to 96%. Studies using ozone gas at various concentrations have shown that ozone is up to 100% effective in inactivating mould fungi in stored cereals (rice, maize, wheat and barley). Research has shown that ozone gas at different concentrations can be used to achieve up to 100% ozone efficiency for the inactivation of mould fungi in stored grains (rice, maize, wheat and barley) [22,23,24,25,26,27,28,29]. According to the results of the studies, it can be observed that, in most cases, the efficiency of ozone gas use depends on the ozone concentration and the exposure time. Also, Sivaranjani et al. [11] point out that the ozone effect may not immediately control the microbial population, but the total number of plaques may decrease during the storage period after ozonation. As mentioned above, ozone gas is used more for grain storage and warehousing, and ozonated water is not suitable for storage because the moisture content of the grain during storage must be below 13–14%. In regard to hydrothermal treatment before milling, ozone gas is also used. Thus, in our study, too, it was found that the use of ozonated water during hydrothermal treatment removes up to 93.6% of mould fungi from the surface of the grain, thereby reducing contamination in the subsequent production process.

### 3.2. Evaluation of the Quality Parameters of Winter Wheat Cereals

Discussing our study alongside studies conducted by other researchers, it can be noted that according to Obadi et al. [12], ozone increases water absorption. It is also suggested by researchers Savi et al. [17] that the presence of moisture plays an important role in the reactivity of ozone with grains because ozone dissolves in water, and this increases the reactivity of the gas with the grains. Moreover, the moisture content is considered as a determining factor for ozone treatment and it is also important to have a longer exposure time at a low ozone concentration rather than a shorter exposure time at a high ozone concentration [11]. Additionally, it is well known that an increase in the moisture content of a product creates a favourable environment for the growth of microorganisms. And different microorganisms have different sensitivities to ozone treatment, which depends on various factors, and the selection of the optimum concentration must be ensured without adversely affecting the product quality [1]. Ozone-induced changes in the flour properties are thought to correlate with structural changes in the two main wheat components: protein and starch [1,30,31]. According to Gozé et al. [30], ozone, by significantly affecting the molecular properties of wheat grain prolamins, substantially alters the rheological properties (i.e., increases the resistance and significantly limits the stretch) of the resulting flour/dough. Many studies have been carried out on starch powders using both gaseous and aqueous ozone and have discussed starch oxidation, where it was found that extensive oxidation reduced the molecular weight by breaking the glycosidic bonds, and consequently could not maintain the integrity of the starch pellets [32,33,34,35,36,37,38]. According to Castanha et al. [39], ozone treatment reduces starch retrogradation and increases the paste clarity. And in the studies by Li et al. [31] evaluating ozone-treated flour starch for use in pasta production, increased pasta dough stability and starch viscosity were observed, and the pasta was firmer, more elastic and less sticky. However, according to studies by Gozé et al. [40] and Sandhu et al. [35], ozone treatment did not cause such significant changes in the physicochemical structure of the starch. In our studies, also no significant changes in the starch were observed as there was no significant change between the ozonated and non-ozonated grains.

Protein, like starch, is a major component of cereals, and this property determines the quality of the final products [41]. Studies show that ozone exposure varies depending on the time and concentration of exposure. For example, the ozone treatment of whole meal cereals at a concentration of 5 g h^−1^ (exposure times of 0, 5, 15, 35 and 45 min) showed a significant increase in the absorption and solubility of water and oil with the increasing exposure time, while showing a significant decrease in the values of the maximum viscosity, decomposition, final and weak viscosity, and also affecting protein and lipids [12]. According to the studies of Obadi et al. [41], prolonged ozone treatment (5 g h^−1^, 1 h) can induce the molecular degradation of proteins and can also affect the reduction in the gluten band intensity. Lee et al. [38] found that ozone treatment did not affect the molecular size distribution of wheat flour proteins. Violleau et al. [42] showed that low ozone treatment increases the ratio of non-extractable polymeric proteins to extractable proteins in wheat kernels, whereas high treatment decreases it. According to the studies of Chittrakorn et al. [43], Wang et al. [44] and Mei et al. [45], the treatment did not affect the protein content of flour. This is in line with our study, where it was found that ozonated water did not affect the protein content of wheat.

In the case of studies where ozone has been investigated for its effect on the grain gluten content, Mei et al. [45] showed that increasing the ozone treatment time to 1.5 h increased the wet gluten content, while further increasing the treatment time to 2 h decreased it. According to the studies of Li et al. [46], we can see that the application of ozone for 4 h at a concentration of 90 mg L^−1^ increases the gluten indenter, the wheat flour dough residence time increases and the protein and sedimentation values are unchanged. Another study shows that the treatment increased the wheat flour falling time (from 445 to 537 s) [47]. In the case of our study, the sedimentation rate was also higher for the grains that were treated with ozonated water than for the grains that were treated with water.

Looking at the overall qualitative changes in cereals, the results of the researchers’ studies show that the response to ozone treatment and its benefits vary widely. In most cases, the studies show either a stable or little impact on the grain quality changes when ozone is used as a preventive measure. This is confirmed by researchers such as Mendez et al. [20], Dubois et al. [14], Dubois, Michel et al. [15], Rozado et al. [16], Tiwari et al. [10] and Freitas, Romenique da Silva de et al. [48], who found that ozone (gas) has minimal to no impact on the quality of cereal crops. Scientists declare that ozone treatment is effective against pathogens and microorganisms, while ensuring adequate or improved product quality [1,49,50,51]. The only decrease in quality occurs when ozone is applied to the products by increasing the amount and the retention time. For example, Mendez et al. [20] observed that ozonated (50 ppm, 30 days) rice hulls changed colour and acquired a vinegar smell that was uncharacteristic of the control samples. Obadi et al. [12] found that ozone gas at a flow rate of 5 g h^−1^ and with increasing retention reduced the qualitative parameters. Scientist Zhu [18] confirms this and states that ozone has clear potential for improving the functionality of cereal products while ensuring food safety. In our case, the wheat studied was Class I, so that even after the application of ozonated water for humidification, these indicators remained unchanged compared to those of Class I.

Therefore, by using ozonated water (1.51 ± 0.1 mg L^−1^) rationally during the hydrothermal treatment stage of winter wheat grains, it is possible to significantly reduce the contamination of products by micromyocetes without exerting a negative impact on the product.

As these studies greatly captivate our interest, we intend to maintain further collaboration with the current company and conduct research in a large-scale production setting in the near future. Additionally, we are highly intrigued by the prospect of conducting life cycle analysis.

## 4. Materials and Methods

### 4.1. Materials

The object of this study is winter wheat grain. Experimental studies with winter wheat grains were carried out in the laboratory of Vytautas Magnus University, Academy of Agriculture, Faculty of Engineering, Department of Mechanical, Energy and Biotechnology Engineering and in a company engaged in grain purchasing, storage and processing, flour production, etc.

The aim was to identify a new method and investigate its effectiveness in moistening grains with ozonated water. In the study, ozonated water was used to moisten the grain at a concentration of 1.51 ± 0.1 mg L^−1^. Figure 4 represents the test scheme for the evaluation of the hydrothermal treatment of winter wheat grain with ozonated water.

Once the initial moisture content of the grain had been determined, the amount of water needed to wet the grain was determined. The prepared grain samples were stored at the specified temperature. The qualitative parameters of the cereals were determined, and the data were evaluated. The detailed methodology for the preparation of the cereals is described below.

### 4.2. Determination of the Moisture Content of Winter Wheat Grains

Before milling, the grain is subjected to a hydrothermal treatment, i.e., after pre-cleaning, before the wheat enters the mill, the grain is humidified with a calculated amount of water to increase the moisture content of the wheat to 14–15% (*a_w_*—0.68–0.70), depending on the initial moisture content [52]. Grains moistened with water are kept for 14 h at a temperature of 30 °C. This temperature is favourable for the growth of micro-organisms.

Before humidification, the moisture content of the grain and the amount of water needed to wet it are determined. The initial moisture content of the grains, the moisture content of the grains wetted with water and ozonated water are determined by drying the grain samples in a Memmert drying oven (Memmert pharmaceutical industry, Schwabach, Germany) at 105 °C to a constant weight.

The grain samples are placed in pre-weighed drying containers (Digital Analytical Scale SCALTEC SPO 51 balance/Scaltec Instruments, Heiligenstadt, Germany). Once the weight of the grain containers has been determined, they are placed in a drying cabinet for drying. After drying, the evaporated water content is calculated:(1)$$m_{v}=m_{i}^{d}-m_{i}^{s},$$
where *m_v_* is the amount of water evaporated from the grain during drying, in g; $m_{i}^{d}$ is the mass of the drying bowl containing the wet grain, in g; $m_{i}^{s}$ is the mass of the drying bowl containing the dry grain, in g.

The moisture content of the grain is then calculated:(2)$$w=\frac{m_{v}}{m_{i}^{d}-m_{i}^{}}\cdot 100,$$
where *w* is the moisture content of the grain, %; *m_i_* is the mass of the empty weighing container, g.

***Determining the water content.*** Once the initial moisture content of the grain has been determined, the amount of water that will be needed to moisten the grain is determined. The amount of water required to moisten the grain is calculated as follows:(3)$$m_{H_{2}O}=m_{d}-m_{0},$$
(4)$$m_{H_{2}O}=\left[\frac{m_{0}\left(100-w_{0}\right)}{\left(100-w_{d}\right)}\right]-m_{0},$$
where *m*_H2O_ is the quantity of water required to wet the grain to the desired moisture content, in kg; *m*_0_ is the initial (dry) mass of grain, in kg; *w*_0_ is the initial (dry) moisture content of the grain, in %; *m_d_* is the mass of wetted grain, in kg; *w_d_* is the desired moisture content of the wetted grain, in %.

***Preparing ozonated water.*** The water used to moisten the grain is potable tap water (hereinafter ‘water’) at a temperature of 15.4 ± 1.3 °C (Thermo E4 thermosensor and Almemo 2590 data logger/Ahlborn Almemo, Holzkirchen, Germany) and a pH of 6.0 to 7.0 (Frisenette pH Test Strips 0–14/Macherey-Nagel, Darmstadt, Germany). The water temperature and pH were tested each time the grain was humidified, with 3 replicates each.

The scheme of the experimental procedure for the humidification of wheat grains (Figure 5(2)) presents the ozonated water preparation system (implemented into a commonly used technology Figure 5(1)). The ozone produced by the ozone generator OZ-AW-15 (power 2.0 kW, water flow 5.0 m^3^ h^−1^, ozone content 15.0 g h^−1^/Ozono centras, Klaipėda district, Lithuania) is directed through a tube with an ejector into a closed water tank, where the ozone is distributed more evenly in the water with mixing paddles. The ozone concentration of 1.5–1.6 mg L^−1^ in the water (5.0 L) is achieved by ozonating for 2.5 min.

The concentration of ozonated water used for grain humidification was 1.51 ± 0.1 mg L^−1^. This concentration was chosen based on scientific research by other researchers that shows that 1.50 mg L^−1^ [53] of ozone in water is sufficient to inactivate microorganisms and that ozone is a highly effective disinfectant at water concentrations ranging between 0.50 and 2.00 ppm (1.00 ppm = 1.00 mg L^−1^) [54]. The Ozone Meter colorimeter Palintest (Palintest, Gateshead, UK) was used to determine the ozone concentration in the water with a measurement range of 0–3.0 mg L^−1^.

Immediately after treatment, one sample is taken from each sample and set aside for the determination of surface micromycete contamination of the products (including untreated control samples).

***Sample preparation and storage.*** Prepare 2 samples with a 0.5 kg sample of grain. The first sample (Figure 4(1)) is moistened with water, the second sample (Figure 4(2)) is moistened with ozonated water. For grain humidification, the supplied water (Figure 5(1)) or prepared ozonated water (Figure 5(2)) is directed to the grain container and sprayed on the grain. The paddles at the bottom of the tank continuously stir the grain layer so that the water or ozonized water is evenly distributed throughout the grain layer. Both samples are stored for 14 h at 30 °C.

***Flour milling.*** To determine the effect of ozonated water on the micromycete contamination of flour, milling of cereals into flour was carried out. Wheat grains were milled by hand (150 rpm) in a domestic mill (LUBA GmbH, Frankfurt, Bad Homburg, Germany). Milling was carried out for whole-grain parts. For the micromycete contamination tests, 3 moistened samples were randomly selected from the total samples.

### 4.3. Assessment of Micromycete Contamination of Winter Wheat Grain and Flour

The micromycete contamination study was carried out in the Microbiology Laboratory of the Institute of Agriculture at the Lithuanian Research Centre for Agriculture and Forestry (Lithuania, Kaunas region). To study the number of colonies of mould fungi on the surface of the winter wheat grain, a study was carried out in selected months (7 months in 1 year). The grains were analysed before and after humidification, after 14 h at 30 °C. Six repetitions were taken during one sampling. In addition, a micromycete contamination test was carried out on ground cereals (flour) for a period of one month.

Mycological contamination testing of winter wheat grain and flour samples were performed in accordance with standards: LST EN ISO 6887-1: 2017 [55]; LST EN ISO 6887-4: 2017 [56]; ISO 21527-2: 2008/P: 2013 [57]. Six repetitions were performed per sampling.

### 4.4. Evaluation of the Quality Characteristics of Winter Wheat Grain

The qualitative parameters of winter wheat grain were determined using the Infratec grain analyser (Foss Tecator AB, Hillerod, Denmark) using infrared spectroscopy technology (LST EN ISO 12099 [58]). The obtained results are compared with the qualitative indicators of wheat, which are given in standard LST 1524:2019 with amendments: “Wheat. Requirements for Purchase and Supply” (Table 1). The wheat quality class were based on the worst indicator threshold value, taking into account only the initial data (before humidification) and the data after hydrothermal treatment (after 14 h). Three replicates were performed per sample (7 months × 3 replicates = 21 replicates).

The quality class of the winter wheat grain is then determined on the basis of the threshold value of the worst indicator, taking into account only the initial data (before humidification) and the data after hydrothermal treatment (after 14 h).

### 4.5. Statistical Evaluation

The data were processed using the IBM SPSS 20 Statistics software (version number 20.0, IBM, Chicago, Illinois, USA), using one-factor analysis of variance (ANOVA with Tukey HSD post-hot test—quite a statistically significant difference), and the level of significance between the data obtained was *p* value of 0.05. The graphs were prepared using the Microsoft Excel software. Data are summarised as means ± standard deviations.

The analysis of the data between the qualitative components of the cereals was carried out using correlation analysis, which compares the *p*-value with the *α*-value to assess the significance of the correlation (significance level *α* = 0.05 and *α* = 0.01).

## 5. Conclusions

The rational use of ozonated water during the milling stage of harvested winter wheat grain can significantly reduce the micromycete contamination of products. In the milling preparation technology (hydrothermal preparation) of winter wheat grains, the use of ozonated water for the pre-milling wetting of the grains—where the ozone concentration in the water is 1.51 ± 0.1 mg L^−1^—reduced the total number of mould colonies by up to 76.4% and 72.8% compared with the water-wetted wheat. In flour, the number of mould fungi after hydrothermal treatment with ozonated water decreased, on average, from 390 ± 110 CFU g^−1^ to 29 ± 12 CFU g^−1^. The treatment of winter wheat kernels with ozonated water had no significant effect on their basic composition.

## Figures and Tables

**Figure 1 plants-12-03267-f001:**
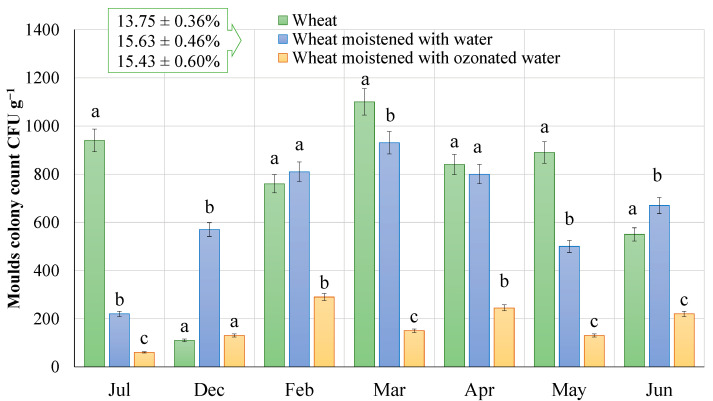
Contamination of winter wheat grain by mould. Results are significant at the *p* < 0.05 level based on one-way ANOVA with Tukey HSD test for month-to-month differences (in letters) (*n* = 6).

**Figure 2 plants-12-03267-f002:**
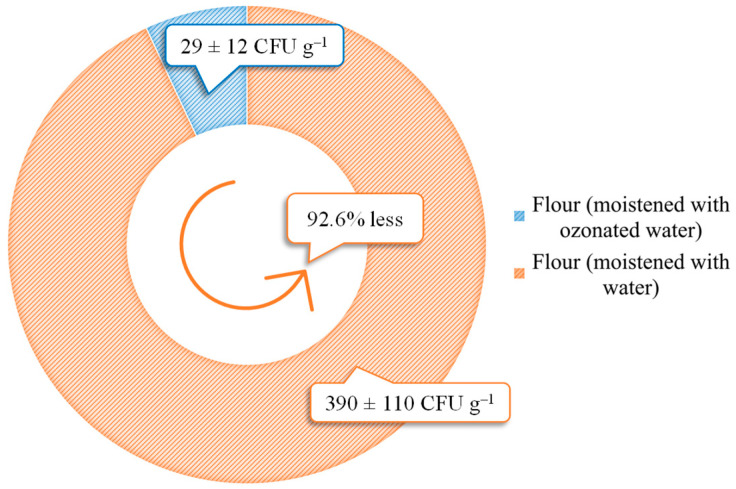
Contamination of winter wheat grain flour with mould (*n* = 6).

**Figure 3 plants-12-03267-f003:**
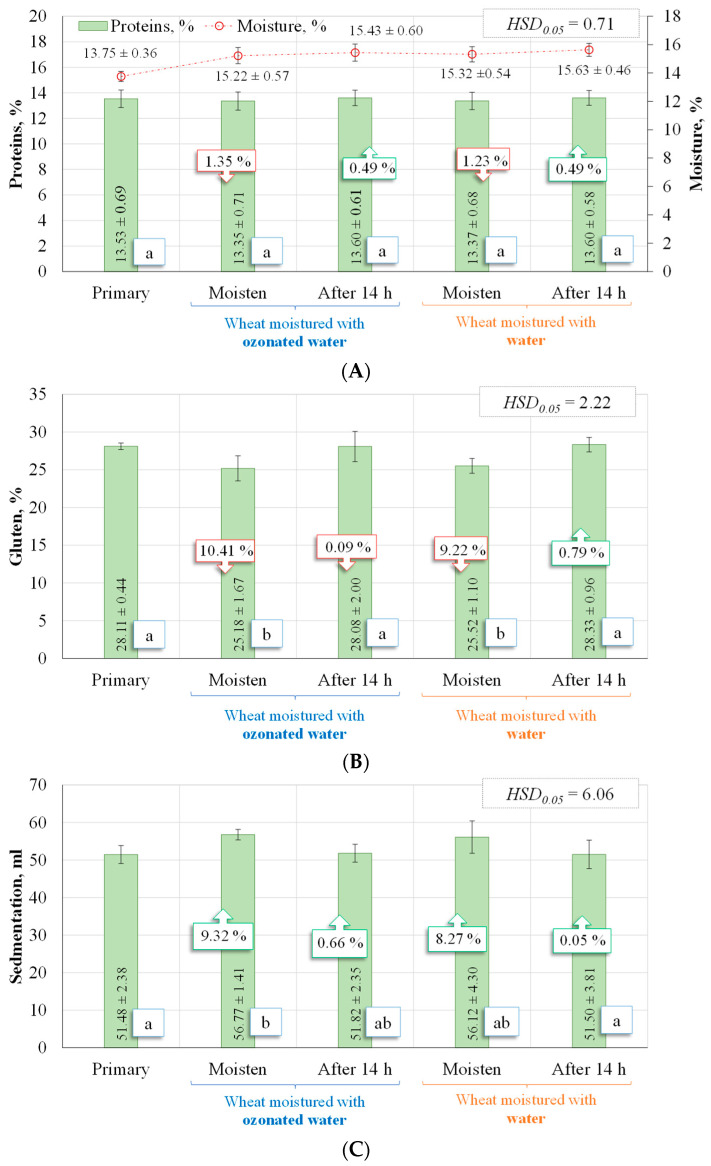
Indicators of winter wheat grains: (**A**)—protein, (**B**)—gluten, (**C**)—sedimentation, (**D**)—starch and (**E**)—hectolitre mass. Results are significant at *p* < 0.05 based on one-way ANOVA with Tukey HSD test (*n* = 21).

**Figure 4 plants-12-03267-f004:**
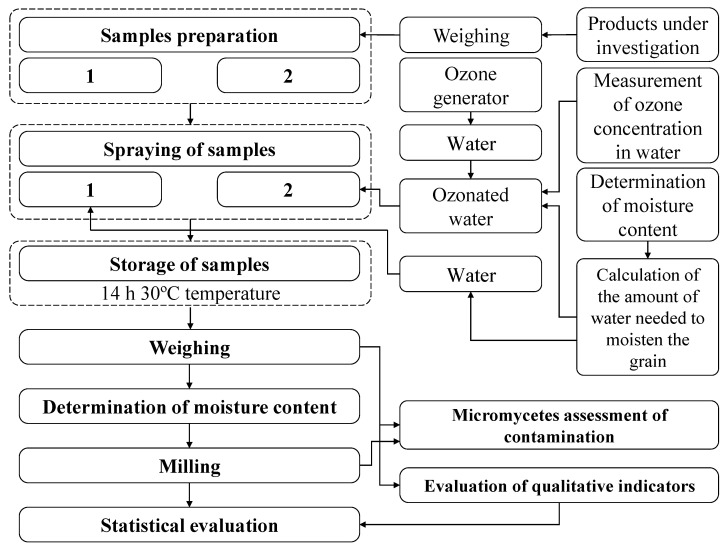
The test scheme for evaluation process of hydrothermal winter wheat grain treatment with ozonated water: (1) The sample was treated with water; (2) The sample was treated with ozonated water.

**Figure 5 plants-12-03267-f005:**
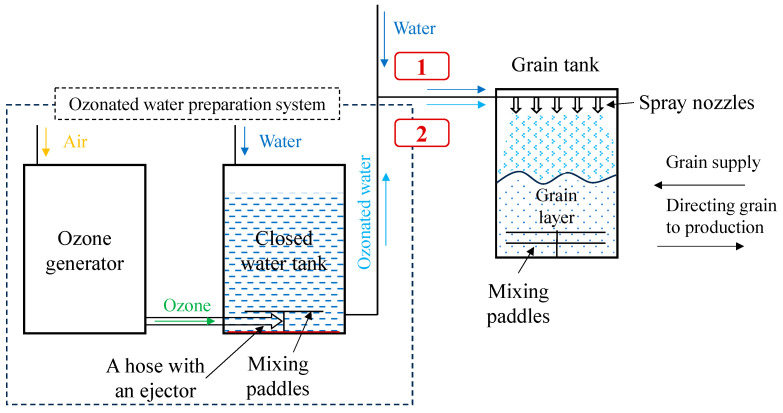
Scheme of the experimental procedure for the humidification of winter wheat grains: (1) The sample was treated with water; (2) The sample was treated with ozonated water.

**Table 1 plants-12-03267-t001:** Quality indicators of wheat (Quality requirements for purchased and supplied according to LST 1524:2019 with amendments) [21].

Parameter	Unit of Measurement	Indicator Values	
		Class I	Class II
Moisture, not more than	%	14.0	14.0
Protein content in the dry matter, not less than	%	13.0	11.5
Sedimentation rate, not less than	mL	35.0	25.0
Gluten, not less than	%	28.0	23.0
Starch	%	65.0–75.0	65.0–75.0
Mass per hectolitre, not less than	kg hL^−1^	73.0	73.0

## Data Availability

The data presented in this study are available in the article.

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
