# Peer review of "Evaluation of Hydrothermal Treatment of Winter Wheat Grain with Ozonated Water"

_plants, 2023, doi:10.3390/plants12183267_

Round 1
Reviewer 1 Report
Feedback on the article: “Evaluation of Hydrothermal Treatment of Winter Wheat Grain with Ozonated Water”
Abstract
Comment 2: The objectives of the study are not clear. Did the authors assess the microbiological decontamination of wheat grains, wheat flour, or both? I suggest specifying that in the abstract.
Keywords:
Comment 1: These keywords were used in the title: ozonated water, hydrothermal treatment, wheat grain.
Introduction
Comment 1: The objectives of the study were not stated.
Materials and methods
Comment 1: All equipment mentioned in the methodology section must be declared respecting the following order: model (manufacturer, city, country). Lines 105; 106; 127-129; 134; 144; 159.
Comment 2: The authors must specify the dimensions of the prototype used in the wheat grain humidification process. A schematic diagram of the experimental procedure has to be provided, depicting the ozone generator, oxygen source, prototype for wheat grain humidification, and details of the system of ozone incorporation into water. What was the oxygen source used for ozone generation? The volumetric flow rate at which ozone gas was incorporated into the water has to be stated. Lines 129-130.
Comment 3: The study employed a concentration of ozone dissolved in water of 1.5 mg L-1. Why did the authors choose this concentration? Line 133.
Comment 4: The water temperature used in the experiment was 15.4 °C. However, the methodology does not mention how it was kept constant throughout the processes. Line 127.
Comment 5: How did the authors humidify the samples? Lines 139-141.
Comment 6: The title of section 2.5 is incorrect. Instead of “Determination of the moisture content of winter wheat grains,” it should be “Statistical analysis.” In this segment, the authors must also inform the experimental design they adopted in the experiment.
Results
Comment 1: Lines 181-193 and 245-249 describe the methodology, but no result was presented.
Comment 2: In lines 365-366, it is unclear what the authors wanted to convey about the study results.
Comment 3: The authors must appropriately discuss the result described in lines 208-211.
Discussion
Comment 1: The authors should provide some suggestions for future studies at the end of the discussion section.
Author Response
Dear Reviewer,
First of all we would like to thank the Reviewer for comments and suggestions, detailed recommendations and contributions to improving the quality of this manuscript “Evaluation of Hydrothermal Treatment of Winter Wheat Grain with Ozonated Water”. All of the issues raised in the Reviewer comments were corrected or commented on in more detail and are listed below. All corrections and answers are highlighted in the attached document.
Sincerely
Simona

Reviewer 2 Report
My main issues with this manuscript are:
- there is no mention of the standard acceptable level for mould/micromyocete on grain so are the values measured in this study of concern?
- the authors assume that micromyocetes are inherently unsafe to health (line 19-20) but there are no data to indicate what species are present, so this risk is overstated
- the level of ozone in the water is initially 1.5mg/L but how long does this persist? In my experience ozonated water is quickly deactivated by contact with complex organic materials - such as grains. Thus, the exposure to water that had been ozonated may have been 14 hours but the ozone may only have been at that level for a matter of minutes, which (according to line 363-364) is probably of little consequence
- the moistened grain is handmilled rather than going through a commercial milling process, so how relevant are the results to commercial practice? What if the grain is milled into a refined flour rather than a wholemeal/wholegrain flour? i.e. what is the effect or removing the bran from the flour on mould levels?
- The Results section contains too much repetition from the Methods section
- the way the Figures are shown is distracting. Fig 3 is not an appropriate summary of the information. Figure 4 needlessly repeats the moisture data and the % change values are not necessary.
- The correlation data in Table 2 are meaningless as there is no comparison of the two water types.
- the word "irrigated" is not correct in this context. Irrigation usually indicates how a crop is watered in a field or glasshouse.
- I do not see the relevance of the State standard for grain quality in this work (Table 1) since none of the quality parameters are sufficiently changed to violate them.
- There is no logical reason to consider data from the grains immediately after moistening because it has no relevance to what happens later. The only valuable information is between water and ozonated water at t=14 hours. So that means a lot of the information/word count in this manuscript is of no value.
- I believe there is no reason for doing the analysis shown in Figure 5 because the ANOVA approach shown in Figure 4 clearly indicates there are no significant differences between at t=14 hours.
The first sentence of the abstract is confusing "Cereals have significantly lower settling losses than succulent agricultural products" - there is no indication of what settling losses are, and what 'succulent' products are you comparing them to?
Author Response

(The authors gave the same response as above.)

Round 2
Reviewer 2 Report
See my comments in red on your coverletter

Author Response
Dear Reviewer,
First of all, we would like to apologize for misunderstanding the remarks and comments. Thank you very much for the clarification.
Additionally, we would like to thank the Reviewer for comments and suggestions, detailed recommendations and contributions to improving the quality of this manuscript “Evaluation of Hydrothermal Treatment of Winter Wheat Grain with Ozonated Water”.
We hope that this time we understood correctly and managed to properly revise the manuscript. All of the issues raised in the Reviewer comments were corrected or commented on in more detail and are listed below. All corrections and answers are highlighted in the attached document.
Sincerely
Simona and Renata

Round 3
Reviewer 2 Report
The effort of the authors to respond to my comments is appreciated. The manuscript has improved. However I still have a few comments:
- the moisture level of the hydrated grains are 15.4% and 15.6% which is slightly higher than the aim of 14-15% moisture (line 131). Is the extra moisture likely to influence the results (would the mould reduction have been lower or higher if it was closer to 15%?)
- Figure 3A is still confusing me. What are the dotted lines supposed to represent? They cannot be the averages of the 3 different treatment groups. They need to be explained in the legend.
- I find it much easier/more logical to think about reductions in numbers as a % value, not as a fold-reduction. Thus lines 285-286, 297 and Fig 3 could be improved.
- line 255. You write that "in all cases" the mould count with ozonated water is significantly lower than the initial and water humidified grain but this is not correct as the Dec data for initial wheat shows.
- line 305, 307 and 319 you say that various protein values decreased, increased and were marginally changed; but the fact is that there was no statistical difference between the values. So in reality there were no changes to report.
Author Response
Dear Reviewer,
Thank you for your observations and questions and we sincerely hope that we have understood and answered them correctly. Also, we would like to thank the Reviewer for comments and suggestions, detailed recommendations, and contributions to improving the quality of this manuscript.
All of the issues raised in the Reviewer comments were corrected or commented on in more detail and are listed below. All corrections and answers are highlighted in the attached document.
Sincerely
Simona and Renata

Round 4
Reviewer 2 Report
Thank you for modifying the manuscript. I believe it is now a much better paper and I hope you agree.